# An organometallic approach to the synthesis of heteropolycyclic compounds from benzenes

Paolo Siano, Louis A. Diment ⓘ, Daniel J. Siela, Megan N. Ericson, Matt McGraw ⓘ, Benjamin F. Livaudais, Diane A. Dickie ⓘ & W. Dean Harman ⓘ ✉

The emerging field of dearomatization capitalizes on the synthetic potential of aromatic molecules. By using a transition metal to bind to two carbons of a benzene ring, the remaining four carbons are left available for the attachment of various chemical fragments. If these fragments are connected, this process could be a blueprint for synthesizing polycyclic architectures. The objective of this study is to develop a modular approach for creating classes of saturated polycyclic compounds that are currently underrepresented in the landscape of druggable chemical space. Herein, the phenyl group of methylphenylsulfone is coordinated to the tungsten complex {WTp(NO)(PMe₃)}, largely interrupting its aromatic stabilization because of strong metal-to-ligand backbonding. Through the combination of ester enolate and amine addition reactions to the arene carbons, a wide array of chemically diverse polyheterocyclic systems is prepared. The tungsten stereogenic center influences the configurations of 3-5 stereocenters derived from the phenyl carbons.

Natural products and their derivatives continue to play a vital role in the discovery of new medicines, especially for treatments in cancer and infectious diseases[1]. Such structures have well-defined three-dimensional architectures, rich in carbon stereocenters, that are optimized for specific interactions with proteins. Molecules rich in stereogenic centers have a greater ability to optimally fill space in a given receptor binding site, hence such complexity is strongly correlated with clinical success[2]. The overreliance on combinatorial methods featuring C(sp²)-C(sp²) linkages has limited the effectiveness of current molecular libraries[3,4], owing to generally poor binding specificities. This realization has brought a renewed focus on natural product-like motifs[5]. These pseudonatural products, defined as natural product-like fragments, are not readily accessible through biosynthesis[6], and are increasingly being recognized as valuable targets in drug discovery. This is particularly the case for polycyclic structures[7,8], where their incorporation into molecular libraries has been slower than expected[9]. The increased rigidity and complex structure of polycyclic compounds make them ideally suited for optimized and selective receptor interactions[1].

The burgeoning field of dearomatization seeks to leverage the synthetic potential of aromatic molecules through a number of diverse approaches, including photoactivation[10,11], high pressures[12], enzymatic protocols[13,14], and activation through transition metals[15,16]. The most common dearomatization strategy using transition metals is to coordinate the arene through all six of its carbons (η⁶)[15,17]. Specifically, coordination to metals such as chromium[18], manganese[19], and iron[19] renders the benzene electron-deficient, thereby promoting the addition of nucleophiles to the aromatic core. While most of these reaction strategies result in substituted benzenes[20], compounds such as cyclohexadienes[21] and cyclohexenones[20] can also be prepared under suitable conditions. It is also possible, however, to bind the benzene substrate through fewer carbons (e.g., η²), enabling the remaining carbons of the ring to participate in conventional alkene reactions[16].

Over the past three decades, our research group has endeavored to develop the chemistry of η²-bound arene complexes of osmium[22], rhenium[23], molybdenum[16], and tungsten[16]. These efforts have led to breakthroughs in understanding how π-basicity affects the stability of the complex and the resulting degree of dearomatization. Notably, the tungsten complex {WTp(NO)(PMe₃)} ([W]; Tp = hydridotris(pyrazolyl)borate) has proven superior in both π-basicity and economic scalability[16,23]. With the metal bound to only two carbons of the

---

Department of Chemistry, University of Virginia, Charlottesville, VA, USA. ✉e-mail: wdh5z@virginia.edu

aromatic ring, the remaining four carbons become available for the addition of various chemical fragments. If such fragments can be linked together, before or after addition to the aromatic ring, the process would establish a blueprint for polycyclic architectures (Fig. 1a, **II, III**)[16]. With the appropriate choices of "bridges", polycyclic structures could be designed to resemble natural products (**IV-VI**) and could find use in diversity-oriented synthesis (DOS)[24–26], biological-oriented synthesis (BiOS)[27], and function-oriented synthesis (FOS)[28–31] approaches to drug discovery.

As mentioned earlier, when a π-basic transition-metal fragment binds to an arene ($\eta^2$), electron density flows from the metal into the arene π* system. Correspondingly, early examples of the synthesis of polycyclic compounds from $\eta^2$-benzenes were initiated by the addition of carbon electrophiles. Typically, these were Michael acceptors[32–34], acetals[35], and less commonly ketenes[35], and carbenes[36], but in all cases, only one cyclization was achieved. In the present study, we explore new methods for the formation of polycyclic cores derived from a benzene core using nucleophilic bridges, primarily using those heteroatoms most commonly found in natural products (N, O, S).

Given the highly electron-rich nature of the transition-metal-bound arene, each nucleophilic addition must be preceded by the addition of an electrophile to reverse the polarization of the metal-ligand interaction. In a recent study utilizing phenylsulfone complexes of tungsten[37], the aromatic ring could accept up to three independent nucleophiles, thereby forming trisubstituted cyclohexene complexes (Fig. 1b, where E[+] = H[+])[37]. These tungsten complexes tolerate exposure to air and water, and can be chromatographed like standard organic compounds. We hypothesized that by tying these nucleophiles together, we could gain access to fused tricyclic or even tetracyclic cores (if "E" is a good leaving group). In this study, we show that such a reaction sequence (Fig. 1C) can provide a highly modular approach to new classes of polycyclic compounds, rich in carbon stereocenters and underrepresented in druggable chemical space.

## Results and discussion
### Cis-fused tricyclics
Previously[38], we demonstrated that protonation of the benzene ligand of WTp(NO)(PMe$_3$)($\eta^2$-benzene) followed by addition of an ester enolate could produce an $\eta^2$-cyclohexadiene complex with a pendant ester group. Further protonation followed by the addition of a primary amine

resulted in a tetrahydroindolone core. We reasoned that if similar chemistry was accessible for a phenylsulfone derivative, then the sulfinate could serve as a leaving group, creating the possibility of additional cyclization (Fig. 2a). Our study commenced with WTp(NO)(PMe$_3$)(3,4-$\eta^2$-PhSO$_2$Me) (**1**), which can be prepared on a multi-gram scale[39]. Diene complexes **2–4** were prepared in good yield via sequential addition of acid followed by an ester enolate[37]. We hoped to carry out the lactam formation with a primary amine connected to a second heteroatom nucleophile that could, in turn, displace the sulfinate group. For our strategy to be successful, the lactamization and subsequent addition of the tethered nucleophile (Nu[3]) needed to occur faster than the competing elimination reaction, as described in Fig. 2b. Gratifyingly, when **2–4** were each combined with 2 eq each of triflic acid (HOTf), excess ethylenediamine, and stirred at −30 °C, the major product formed was a tricyclic (**5–7**, Fig. 2a). Several intermediates were observed by stopping the reaction early (Fig. 3), but if the mixture was allowed to stir a full 72 hours the intermediates gave way to the desired products (**5–7**), provided that during the workup the product mixture was kept under basic conditions (vide infra). This one-pot reaction sequence was shown to be remarkably general as a number of commercially available 1,2-diamines led to a triazaacenaphthylene core (**5–17**). In cases where steric interactions differentiated the two primary amine groups, the less encumbered nitrogen selectively formed the lactam ring (**8–11**). Further, in all cases but **10** and **15**, a single diastereomer was isolated (dr > 20:1) that was shown to be the result of the ester enolate, and both amine additions occurring anti to the sterically demanding metal complex {WTp(NO)(PMe$_3$)}[16]. When the incorporated diamine itself contained additional stereocenters, diastereomers resulted. Specifically, compound **10** was synthesized as a mixture of diastereomers starting from racemic **3** and (**R**)-propane-1,2-diamine. In contrast, compounds **11** and **12** were prepared using enantiopure (R)-**3**, in combination with (R)-propane-1,2-diamine or (S)-propane-1,2-diamine, respectively (vide infra). The ee for compounds **11** and **12** is presumed to be limited by the optical purity of the diamine (ee = 99%).

Ethanolamine also gave satisfactory results (**18**). When 1,3-diamines or 1,4-diamines were used, tricyclics containing seven- and eight-membered rings were formed, respectively (**19–22**). 2D NMR data techniques (COSY, NOESY, HSQC, HMBC) were used to confirm the structure of every compound (Fig. 2a) along with SC-XRD data for complexes **5–16**, and **18–21**. These compounds were observed to be

**a** Objective: polycyclization of benzene via $\eta^2$-TM dearomatization

● metal-controlled stereocenter

X = C, N, O, S

etc.

**examples of 3D structures of type VI cores**

[6/6/6/6]   [6/6/6/5]   [6/6/5/5]

**b** Phenyl sulfone dearomatization

[W] = WTp(NO)(PMe$_3$)

**c** Proposed: nucleophile-derived polycyclizations

**Fig. 1 | Conceptual overview of this study. a** Shows the conceptual approach to synthesis of polycyclic cores using benzene as the nucleus. **b** Shows the dearomatization of phenylsulfone using a tungsten reagent, and **c** applies this chemistry to the synthesis of tricyclic and tetracyclic molecules using "double nucleophile" bridges. Tp = hydridotris(pyrazolyl)borate. Green atoms represent attachment points.

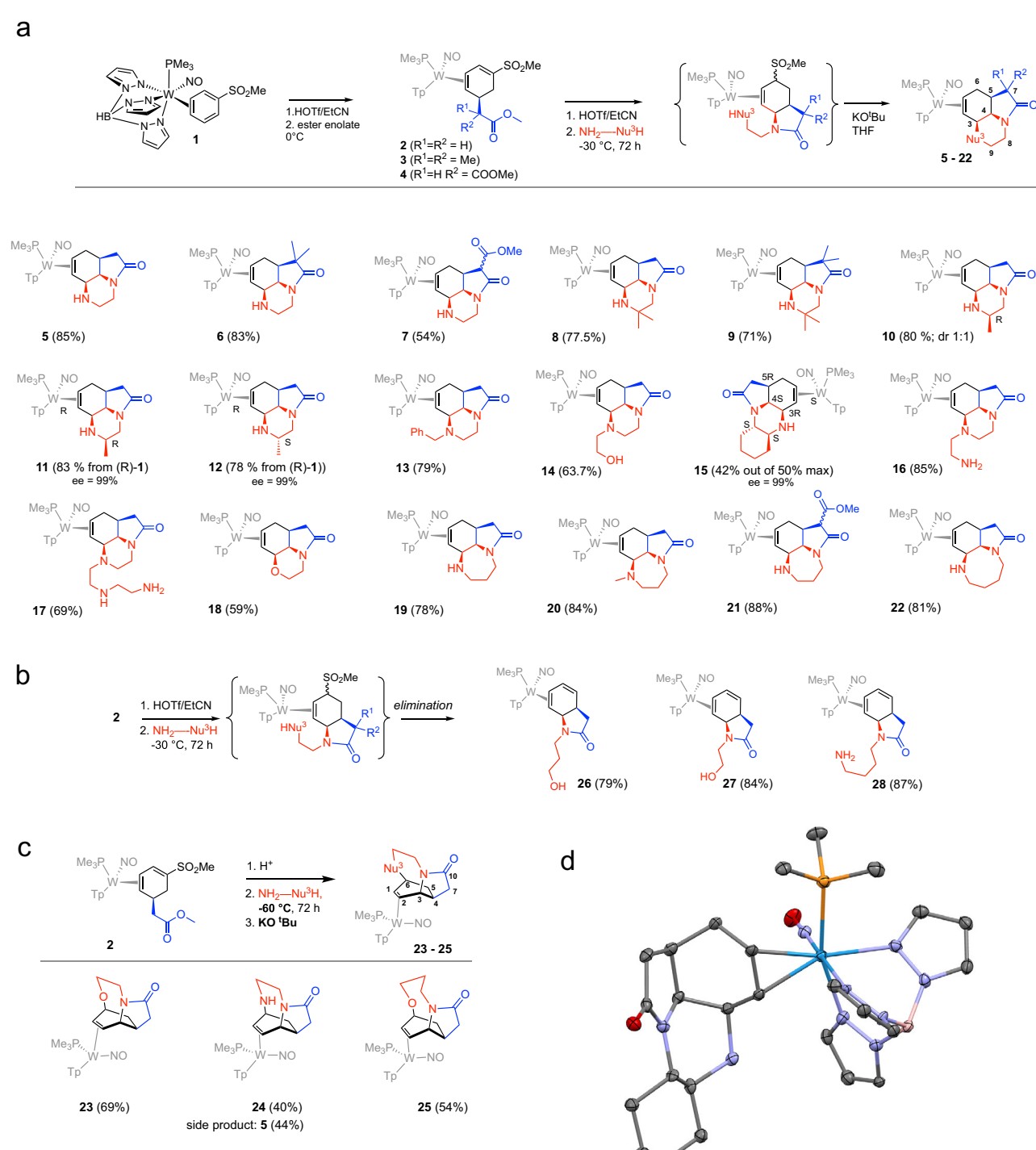

**Fig. 2 | Multicyclic products obtained through diamines. a** Fused tricyclic compounds. **b** Diene lactam complexes are produced as byproducts of tricyclic formation. **c** Bridged tricyclic compounds. The ee for compounds **11, 12**, and **15** is presumed to be limited by the optical purity of the diamine (ee = 99%; Sigma-Aldrich). Blue represents the ester fragment. Red represents the amine fragment. **d** Molecular structure of **15** as determined by SC-XRD (50% ellipsoids). The absolute stereochemistry of **15** was confirmed with a Flack parameter of −0.019(2).

stable as solids, or in a basic solution, but partially underwent a ring-opening reaction under mildly acidic conditions or long periods (3–7 days) in solution at ambient temperature, forming diene lactams analogous to **26–28** (Fig. 2b). Hence cold reaction temperatures and basic workup conditions were paramount in successfully running the reactions to completion with high purity. In the case of HNu₂− Nu₃H = (S,S)-trans-1,2-diaminocyclohexane, *(S_W,3R,4S,5R,8S,9S)*-**15** was isolated via chromatography in 42% yield, this compound appearing in

the methanol fraction. Since **15** was obtained using a racemic form of phenylsulfone complex **1**, only 50% of **1** was available to form this stereoisomer. Of note, the other possible stereoisomer of **15**, *(R_W,3R,4S,5R,8S,9S)*-**15**, was never identified. Instead, the other hand of the metal was isolated as *(R_W,3S,4R,5S,8S,9S)*-**41** (vide infra, Fig. 4), which crystallized (35% yield out of a theoretical yield of 50%) from the low-polarity fractions of the chromatography procedure for **15**. The ee for compounds **15** and **41** is determined by the optical purity of the

**Fig. 3 | Mechanistic representation for the synthesis of tricyclic compounds.** Proposed pathways for the formation of desired polycyclic complexes and undesired diene elimination products. Energies reported are in kcal/mol for the case where Nu³ = NH. Blue represents the ester fragment. Red represents the amine fragment.

diamine, and the Flack parameter for the corresponding crystal structures confirms the absolute stereochemistry shown in Figs. 2 and 4.

## Bridged tricyclics

When the reaction sequence described above was run under colder temperatures (−60 °C, 72 h), a new species was observed (Fig. 2c), typically as a mixture along with the fused tricyclic complexes. In one example (**23**), a single compound could be separated from its tricyclic analog **18**. Spectroscopic features for **23** were similar to those observed for **18**, but NOE data indicate that both allylic carbons of the bound alkene were connected to heteroatoms, with H6 oriented toward the PMe₃ ligand. A homolog of the 2-aminoethanol-derived (**23**) could also be prepared from 3-aminopropanol (**25**) that had no fused tricyclic counterpart observed. Meanwhile, whereas the bridged tricyclic **23** could be prepared as the dominant species at −60 °C, we were unable to separate **24** from its fused tricyclic isomer **5**. A full spectroscopic analysis revealed that this family of compounds **23**–**25** were rare examples of N-C6 bridged indolones. Regrettably, our attempts to grow single crystals of **23** or **24** species resulted in conversion to a mixture containing the fused tricyclic species (e.g., **5** or **18**) and diene complexes (**27** or the analog of **24**).

## Competing mechanisms

The likely mechanism for the formation of the fused and bridged polycyclics starts with the protonation at the terminal position of the diene (C5 of **2**; Fig. 3). The resulting η²-allyl complex exists as two conformations (**IP** and **ID**) differing by the position of the carbenium carbon (C6 for proximal to the PMe₃ (**IP**) or C3 if distal to the PMe₃ (**ID**))[40,41]. According to DFT calculations, the distal form is favored by 4.8 kcal/mol, influencing the amine to add to C3, adjacent to the ester group, thereby inducing the desired lactamization. The steric bulk of the metal complex requires both the addition of the ester enolate and the amine to add anti to the metal[16].

The lactamization places the sulfone group in **IV** in an allylic position. Subsequent metal-stabilized loss of the sulfinate group forms a secondary allyl species (**III**), also present as two conformers. In contrast to the initial allyl (**I**), the distal conformer is roughly equal in

free energy to the proximal form, owing to the electron-withdrawing nature of the amide nitrogen. This is consistent with the formation of an alternative tricyclic species that can be formed at low temperatures, where the tethered nucleophile adds to C6 of **IIIP** rather than C3 of **IIID**. As described earlier, using cold temperatures (−60 °C) and basic reaction conditions (potassium tert-butoxide), we were able to trap and isolate the bridging tricyclics **23**–**25** (Fig. 2c). While the bridged tricyclics may be kinetically competitive, calculations indicate that they are strongly disfavored thermodynamically compared to the fused tricyclics. For example, the ethylenediamine-derived bridged tricyclic **24** is 11.8 kcal less stable than its fused tricyclic isomer **5**. As shown in Fig. 3, in cases where the allyl species **III** fails to evolve into a tricyclic, elimination can occur to form the diene **IX**. Spectroscopic evidence confirms the formation of diene **28** (**IX**, where HNu³ = NH₂) as a decomposition product of **24**, even though DFT calculations comparing structures **VII** and **IX** for Nu³ = NH indicate that the diene isomer **IX** is ~4 kcal/mol less stable than its fused tricyclic isomer (**VII**). Conversely, there is no experimental indication of a purported dienamide isomer of type **V**, even though DFT calculations place such a species about 3 kcal/mol lower in energy than the diene **IX** (for Nu³ = NH). We speculate that this is a kinetic phenomenon in which deprotonation at the bridgehead carbon (C4) is challenging, in part because such a reaction would require a base to approach from the same side as the sterically demanding metal complex. Calculations for the ethylenediamine-derived tricyclic **5** indicate that the purported trans-fused isomer of **5** was only marginally disfavored (0.7 kcal/mol), although there was no sign of this species experimentally. Apparently, the kinetic barrier required for the allyl species **IIID** to bring the tethered amino group to C3, syn to the metal (**VI**) is too high to be competitive with the addition anti to the metal.

## When lactamization fails

Aniline derivatives were also considered as candidates for lactamization and subsequent ring-closure. Encouraged by the reactivity displayed by ethylenediamine, we first explored o-phenylenediamine. Signals in the ¹H-NMR spectrum revealed a mixture of two products, neither of which corresponded to the patterns observed for the fused or bridged tricyclics (Fig. 2a and 2c). In particular, an ester methoxy

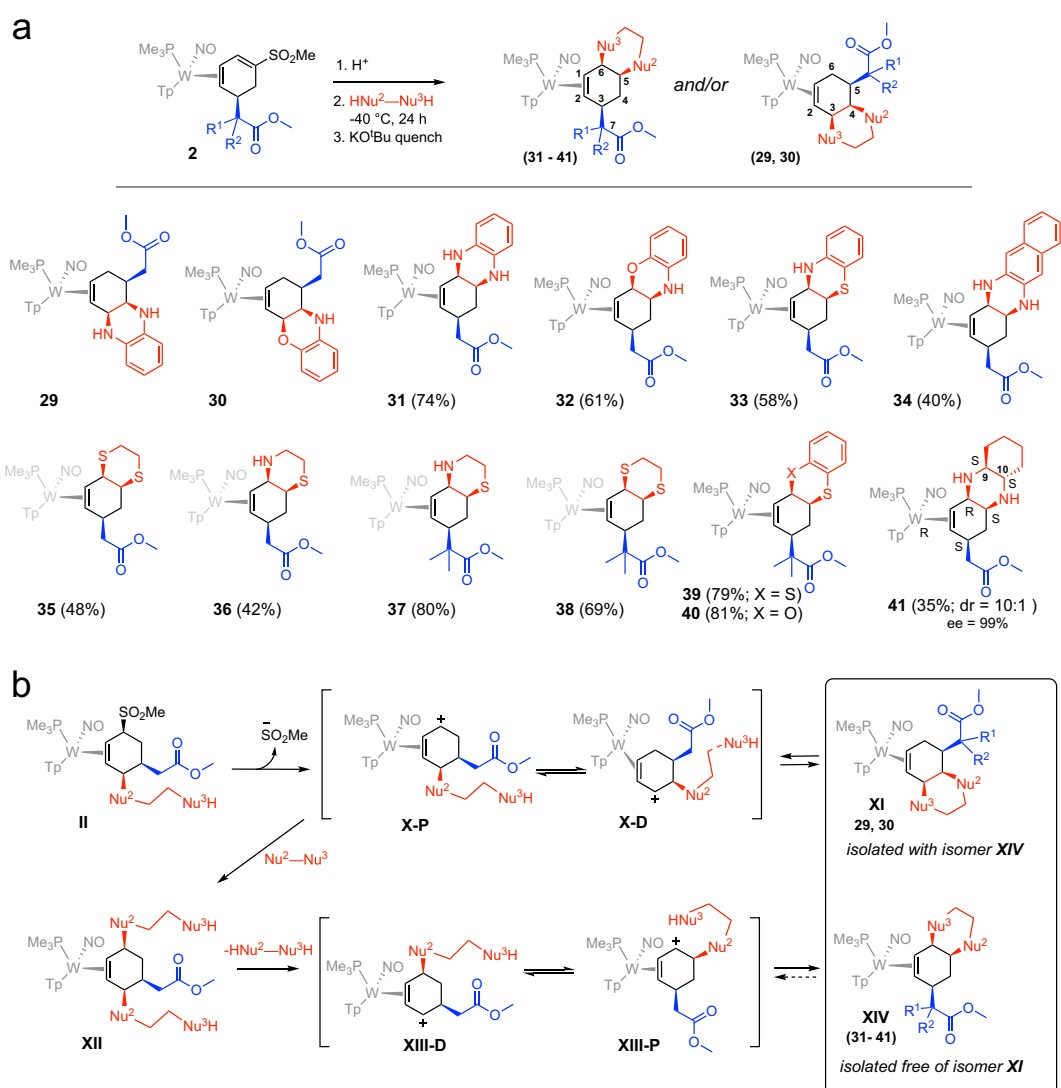

**Fig. 4 | Formation of fused bi- and tricyclic esters. a** A range of heterocycles formed with pendant ester group. **b** Proposed mechanism for the formation of bicyclic compounds **29**–**41**, where loss of sulfinate group occurs prior to lactam formation. The absolute stereochemistry of **41** is confirmed by SC-XRD with a Flack parameter of −0.012(3). The ee for **41** is presumed based on that of the diamine (99%). Blue represents the ester fragment. Red represents Nu$^2$-Nu$^3$ fragment.

signal was present for both compounds. After a thorough evaluation of 2D NMR data, it was posited that the two products formed were **29** and **31** (Fig. 4a), where the cyclization that occurred did not involve a lactamization. The scope for this reaction pattern was expanded using other aniline derivatives, including *o*-aminothiophenol (**33**), *o*-aminophenol (**30**, **32**), and 2,3-diaminonapthalene (**34**), where Nu$^2$ is defined as the first of the two nucleophiles to add. We noticed similar behavior with thiols, including *o*-benzenedithiol (**39**), 2-mercaptophenol (**40**), and aliphatic variants such as ethanedithiol (**35**, **38**), and cysteamine (**36**, **37**). In addition to the methoxy signal, compounds **29**−**41**, show a clear NOE interaction between the PMe$_3$ and the allylic proton H6. Further, a splitting pattern for H1 and H2 is observed to significantly differ from that seen for the fused and bridged tricyclics, indicating a different cyclohexene ring conformation in these bi- and tricyclic esters: XRD data were successfully obtained for compounds **30**, **33**, **34**, **35**, **38**, **39**, **40**, and **41** demonstrating the half-chair nature of the cyclohexene ring as opposed to the boat conformation observed for the fused tricyclics. In particular, the compound ($R_W$,3S,5S,6R,9S,10S)-**41** was separated from other isomers through crystallization in 35% yield (maximum yield of 50%). This compound was obtained as a byproduct of the synthesis of **15**, and its stereochemistry reflects the

fact that most of the S hand of the tungsten was consumed in the formation of **15** (vide supra).

While the reactions forming the ester polycyclics **29**−**41** did not yield the originally desired products, they did provide some additional mechanistic insight. Whereas primary aliphatic amines were successful in forming the desired lactam ring, in the case of arylamines, lactamization is not as favorable, presumably due to the weaker nucleophilicity of the amine. Compounds of the type **II** could only be isolated when the nucleophile was ethanedithiol (SI, **42**). Indeed, loss of the sulfinate occurs readily (**II** → **X-P** Fig. 4b) followed by the addition of a second arylamine (**XII**, Fig. 4b). While allylic amine complexes are stable under basic conditions, in the presence of weak acids (i.e. ammonium salts in the reaction mixture) they are labile[42]; loss of the distal arylamine ligand in **XII** generates allyl **XIII-D**. An allyl shift to **XIII-P** then leads to the ring-closure that generates compounds of type **XIV**. In two cases (**29** and **30**), we observed a competing process in which loss of the sulfinate was immediately followed by a ring-closure to form compounds of type **XI** (formed as a mixture with **XIV**). This cyclization likely occurs for many of these complexes but is reversible. Indeed, whereas **29** could not be purified from **31**, **30** was successfully separated from **32** through recrystallization of **30** in a solution of

acetonitrile. Complexes **31-41** were isolated free of impurities of the form **XI**, but partial decomposition of the reaction mixture lowered the overall yield. When Nu² is an arylamine, the bond between the allylic carbon and Nu² is sufficiently inert that ring-closure can lead to the isolation of compounds of type **XI**. Where Nu² is S, the reaction proceeds to bicyclic compounds of the type **XIV**. An exception to this general trend occurs with cysteamine in the preparation of compounds **36** and **37**, where we speculate that the nitrogen is actually protonated (a zwitterionic form of cysteamine), thereby preventing the ring-closure from occurring to form **XI**. Parenthetically, the lack of an o-phenylenediamine lactam also led us to investigate aniline itself as a potential nucleophile toward lactamization. Consistent with our mechanistic hypothesis that aniline derivatives are not sufficiently nucleophilic to form the lactam (**IV**), the indolone core was not formed when complex **2** was added to aniline (SI, **82**).

### Liberation of organic tricyclics
In an effort to liberate the synthesized polycyclic molecules from the {WTp(NO)(PMe₃)} fragment, several different approaches were employed. In most cases, 1–2 eq of Br₂ proved effective in liberating the organic product (e.g., **43-57** in Fig. 5) with yields ranging from 58 to 73%.

When the organic polycyclic contained sulfur, oxygen, or an aryl amine, this approach was generally unsuccessful, as the effect of the oxidant resulted in loss of the heteroatom substituent and reformation of an η²-allyl tungsten complex. In some of these cases, the use of DDQ was shown to be effective (e.g., **58, 59, 62, 63, 65**) with yields ranging from 57%–74%. Compounds of the type **64** were obtained by stirring the parent complex **32** in acetonitrile over the course of a week. Unfortunately, the bridging tricyclics shown in Fig. 2c, proved elusive. Figure 5a, summarizes the successfully isolated organics.

### Selective formation of complex diastereomers
Given the potential value of these polycyclic cores as precursors for pharmaceutical leads, we sought the ability to prepare specific diastereomers using a pair of enantioenriched fragments. Phenylsulfone complex *(R)*-**1** can be synthesized from an enrichment procedure previously published[42]. We then converted it to *(R)*-**2** and combined this ester with the R and S forms of 1,2-diaminopropane to form compounds **11** and **12**, which were then elaborated into their liberated organics (R,S,S,R)-**48** (77%), and (R,S,S,S)-**49** (72%) (Fig. 5). Proton NMR data confirm the presence of just one diastereomer (dr > 20:1).

### Extensions to tetra- and pentacyclic compounds
Although the emphasis of this study has been on heteroatom nucleophiles (Nu² and Nu³), we recognize that the hydroacenaphthene core is also prominent in natural products. To this end, we sought to demonstrate the methodology described in Fig. 1 where Nu³ was carbon. Selecting 2-(aminomethyl)indole as the bridging-nucleophile, we prepared the indolone **66** from ester **2**, in a similar manner to that shown in Fig. 2a. Upon standing for 1 day at -30 °C, then at room temperature for a day, this species slowly evolved cleanly into the pentacyclic **67**. Stirring **67** in acetonitrile for a week afforded the final organic **68** in 55% yield (Fig. 6a, one-pot from ester **2**).

As discussed earlier, benzene offers a unique scaffold in providing a ring of six unsaturated carbons that could potentially be elaborated into new stereocenters. Shifting our focus to methods that utilize additional benzene carbons, one approach takes advantage of the ability of Selectfluor® to oxidize η²-diene complexes to biallylic ether complexes[36,43]. Given the liability of alkoxy groups at the allylic positions of this tungsten system, such an oxidation could provide additional points of attachment for cyclizations. To demonstrate this, we treated the diene **69** with Selectfluor in the presence of MeOH to generate the tricyclic **71**, an analog of tricyclic **18** with an allylic methoxy group and an additional carbomethoxy tether (Fig. 6b). Exposing this material to acid followed by methylamine provided the [6/5/5/6] tetracyclic bis-lactam **75** in good yield. A similar process delivers the [6/5/5/7] analog **76**. A crystal structure of **75** confirms the all-cis stereochemistry of the tetracyclic ring system. Further reaction with AgNO₃ provides the final tetracyclics, **78** and **79**, in 50% and 36% yield, respectively, and an SC-XRD analysis of the organic **79** confirms its structure. When derivative **80** is prepared with a pendant alcohol, treatment of the free alkene with bromine induces formation of a cyclic ether, resulting in the meso-pentacyclic compound **81** (25%), which we have named *mobulamide* owing to its resemblance to the Mobulidae family of rays. Further extensions of this chemistry are realized with the successful syntheses of analogous polycyclic cores as elaborated from the δ-lactam **84**. An example of a [6/6/6/6] ring system is found with the synthesis of the tetracyclic **85** (Fig. 6c).

### Potential as a tool in drug discovery
The field of dearomatization is growing exponentially[43-45], motivated by the ability to leverage multiple carbons of an aromatic ring in the synthesis of more complex products. Prior to this work, most of the examples that resulted in a new ring involved only one or two carbons of

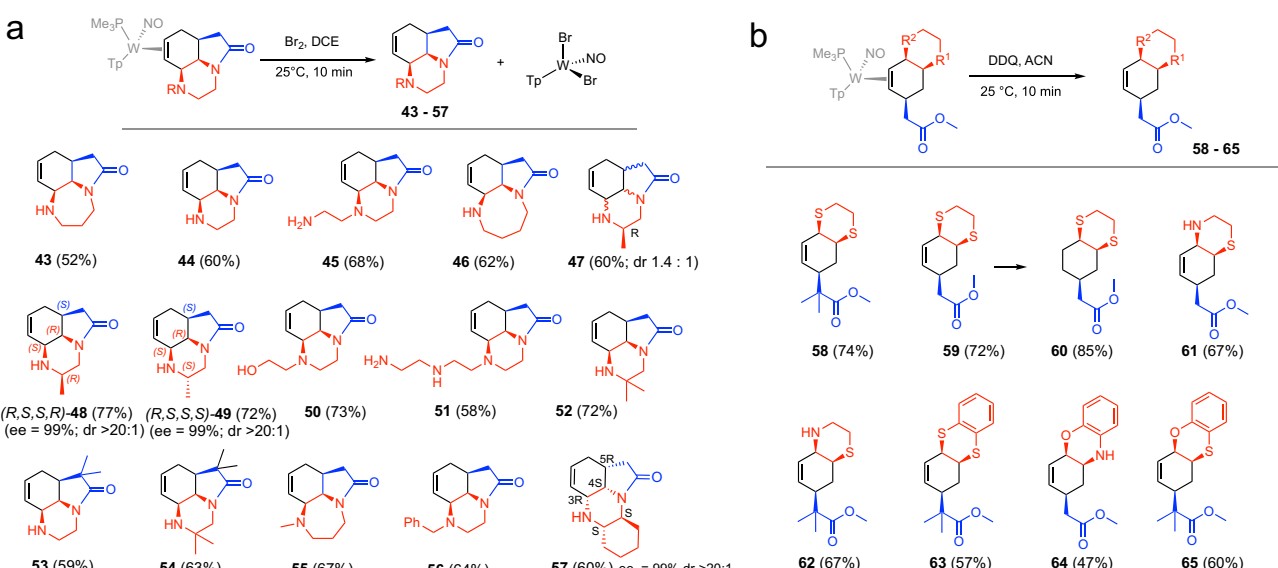

**Fig. 5 | Oxidative decomplexation of heteropolycyclics. a** Oxidation via bromine. **b** Oxidation via DDQ. For compounds **48, 49**, and **57** the ee is assumed on the basis of the ee for the diamine reagent (99%; Sigma-Aldrich). Blue represents the ester fragment. Red represents Nu²-Nu³ fragment.

**Fig. 6 | Preparations of advanced heteropolycyclics. a** The preparation of an indolizidine pentacyclic via 2-(aminomethyl)indole. **b** The preparation of bis-lactam polycyclics prepared from dimethylmalonate. **c** Further cyclizations based on the tungsten-protected cyclohexene. Blue represents the ester fragment. Red represents Nu²-Nu³ fragment. Orange represents the third-stage amine. SC-XRD drawings displayed at 50% ellipsoids.

the aromatic substrate. In contrast, polycyclizations of benzenes are virtually absent in the literature. One exception is the recent synthesis of the pentacyclic core of the lycorine family carried out with tungsten, but this still represents only one new ring formed while the arene was bound to the metal[42]. Other examples of more elaborate cyclizations with aromatics include the photocyclization of ω-phenylalkene derivatives in the synthesis of penifulven C[46] and ceratopicanol[47]. These examples involve carbon-carbon linkages rather than those with heteroatoms.

Many of the top-selling pharmaceuticals contain tri-, or tetracyclic molecules with carbon stereocenters incorporated in the polycyclic core[46]. Examples include steroids (e.g., Symbicort, Trelegy, Relvar, Zytiga, Seretide, Mirena, Flixotide, Pulmicort, Premarin, Avodart, Breztri, Faslodex, and Medrol), tri- and tetracyclics incorporating nitrogen heterocycles (e.g., Biktarvy, Tivicay, Triumeq, Latuda, Dovato, Ingrezza, Austedo, Juluca, Cialis, Cabenuva), tetracycline antibiotics, (e.g., Vibramycin, Arestin, Xerava, Seysara, Nuzyra), taxol derivatives (e.g., Abraxane, Jevtana), and tri- and tetracyclic anti-depressants (e.g., Ludiomil, Tolvan, Remeron, Coaxil). Other appearances of tricyclic nitrogen cores include the stemona[48] and lycorine[49] alkaloid families, many of which include the all-cis [6/6/5] core[49] analogous to **44** or the all-cis-[6/7/5] core[48] analogous to **43**. Yet remarkably, synthetic routes to the [6/6/5], [7/6/5], [8/6/5], [6/6/6], [6/6/5/5], [6/7/5/5], [6/6/6/5], and [6/6/6/6] cyclic skeletons (CSK)[9] (regardless of atom type) shown in Figs. 5 and 6 with all-cis ring junctures are practically absent in the chemical literature, likely owing in part to the lack of general methods to prepare them. An important exception is the lycorine family of compounds, for which numerous synthetic approaches have been reported. In particular, Baudoin et al have

demonstrated an attractive synthesis of γ-lycorane in which a benzene precursor was hydrogenated to generate the required all-cis stereochemistry[50]. Over- and mis-hydrogenation were competing reactions, and the synthesis of the aromatic precursor required a challenging double Pd-catalyzed C(sp²)-H/C(sp³)-H arylation. In comparison, the one-pot double cyclization methodology described herein is more modular and does not rely on precious metal catalysts. Specifically, we recently synthesized γ-lycorane from the tungsten benzene complex in six steps (25% overall yield)[42]. However, the polycyclic cores that have not yet been prepared in a lab or found in nature are the most exciting potential targets for this methodology.

Like γ-lycorane, tricyclic compounds (**43**–**56**) and tetracyclic **57** in Fig. 5 and polycyclics **68**, **78**–**81**, **85** in Fig. 6 meet all the criteria for Lipinski's rule of five[51] for evaluating drug likeness, as well as the criteria of Ghose[52], Veber[53], Egan[54], and Muegge[55], and are in a molecular weight range that is ideal for fragment libraries[56,57]. Even the cis-fused bicyclics in Fig. 5 (e.g., **58** - **65**) are not well represented (only 11 examples of cis-1,4-thiaazadecalins and 16 examples of cis-1,4-dithiadecalins). We believe that the current lack of conventional synthetic methods combined with the predicted bioavailability and drug likeness of the compounds in Figs. 5 and 6 makes a strong case for this methodology providing access to completely unexplored regions of chemical space where there is a high chance of identifying lead compounds.

## Methods
NMR spectra were obtained on 400, 600, or 800 MHz spectrometers. Chemical shifts are referenced to tetramethylsilane utilizing residual ¹H signals of the deuterated solvents as internal standards. Chemicals

hifts are reported in ppm, and coupling constants (J) are reported in hertz (Hz). Infrared spectra (IR) were recorded as a solid on a spectrometer with an ATR crystal accessory, and peaks are reported in cm⁻¹. Electrochemical experiments were performed under a nitrogen atmosphere. Most cyclic voltammetric data were recorded at ambient temperature at 100 mV/s, unless otherwise noted, with a standard three-electrode cell from +1.8 to −1.8 V with a platinum working electrode, acetonitrile or N, N-dimethylacetamide solvent, and tetra-butylammonium hexafluorophosphate (TBAH) electrolyte (-1.0 M). All potentials are reported versus the normal hydrogen electrode (NHE) using cobaltocenium hexafluorophosphate ($E_{1/2}$ = −0.78 V, −1.75 V) or ferrocene ($E_{1/2}$ = 0.55 V) as an internal standard. The peak separation of all reversible couples was less than 100 mV. All synthetic reactions were performed in a glovebox under a dry nitrogen atmosphere unless otherwise noted. All solvents were purged with nitrogen prior to use. Deuterated solvents were used as received from Cambridge Isotopes and were purged with nitrogen under an inert atmosphere. When possible, pyrazole protons of the tris(pyrazolyl)borate (Tp) ligand were uniquely assigned (e.g., "Tp3B") using two-dimensional NMR data. If unambiguous assignments were not possible, Tp protons were labeled as "Tp3/5 or Tp4". All J values for Tp protons are 2(± 0.4) Hz. BH peaks (around 4−5 ppm) in the ¹H-NMR spectra are not assigned due to their quadrupole broadening; However, confirmation of the BH group is provided by IR data (ca 2500 cm⁻¹). Full characterization of compounds is provided in the SI. Ground-state structures were optimized at the M06 level of theory using the 6-31 G**[LANL2DZ for W] basis set in Gaussian 16.

## Data availability

The synthetic, spectroscopic, and electrochemical data generated in this study are provided in the Supplementary Information file. CCDC deposition numbers 2432194-2432229 contain the supplementary crystallographic data for this paper. These data can be obtained free of charge from The Cambridge Crystallographic Data Center via www.ccdc.cam.ac.uk/structures. Source data are provided with this paper (DFT calculations). All data are available from the corresponding author upon request. Source data are provided with this paper.

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

## Acknowledgements

Research reported in this publication was supported by the NIGMS of the National Institutes of Health under award numbers **R01GM132205** (W.D.H.) and **R35GM152065** (W.D.H.). Single crystal X-ray diffraction experiments were performed on a diffractometer at the University of Virginia, funded by the NSF-MRI grant **CHE-2018870** (D.A.D.). Some NMR data were collected on an instrument funded by the NSF-MRI grant **CHE- 2215062** (W.D.H.). The content is solely the responsibility of the authors and does not necessarily represent the official views of the National Institutes of Health or the University of Virginia.

## Author contributions

W.D.H. and P.S. conceived the original project. W.D.H., P.S., L.A.D., D.J.S., M. M. and B.F.L. designed experiments, prepared samples, and collected NMR and HRMS data. D.A.D., P.S., L.A.D. and D.J.S. carried out X-ray molecular structure determinations. M.N.E. carried out DFT calculations. W.D.H., P. S., D.J.S. and L.A.D. wrote the manuscript.

## Competing interests

The authors declare no competing interests.
