## [Transparent Peer Review file · Nature Communications]

An organometallic approach to the synthesis of heteropolycyclic compounds from benzenes

Corresponding Author: Professor Walter Harman

Version 0:

Reviewer comments:

Reviewer #1

(Remarks to the Author)

The manuscript by Harman and coworkers reports a new strategy for the synthesis of heteropolycyclic molecules from phenylsulfone which is coordinated to a tungsten complex. This dearomatization allows for a stepwise functionalization. Finally the tungsten fragment is oxidatively removed. This is a refreshing and elegant way to derive at complicated polycyclic structures which are natural-product-like and therefore attractive for drug design and discovery. I like this approach and think that it is suitable for publication in Nature Communications.

A few points should be addressed:

- Abstract and Summary should be improved: It needs to include some more information about the new method.
- Introduction or first part of the results section: A comment on the reason why tungsten is especially suitable for this approach would be useful.
- Most reader will not be familiar with the employed tungsten- η^2 -benzene complexes: Can these complexes be chromatographed like standard organic compounds? A remark would be useful.
- Figure 2: Reaction conditions for the ester enolate addition should be included.
- Aspects of regioselectivity and diastereoselectivity are not discussed sufficiently (except for mentioning diastereoselectivity for cpds 48 and 49). This is an important point that should be addressed.
- Compounds 48 and 49: The authors state that proton NMR data confirm the presence of just one diastereomer. But what is their enantiomeric excess?
- Mechanistic Figure 3: The authors write that the allyl complexes IP and ID differ in their their conformations. That appears to be wrong. IP and ID are not conformers but instead may be called regioisomers. By the way: Is it possible that IP and ID are not in equilibrium but rather resonance structures and exist as an η^3 -allyl complex? Same for IIID and IIIP.

Reviewer #2

(Remarks to the Author)

This article from Siano et al. reports on the derivatization of phenylsulfone coordinated in an η^2 fashion to a Tp-supported nitrosyl tungsten complex. Binding to tungsten ensure sufficient electron-richness to the arene so that it can be easily protonated, the entry point for dearomatization and functionalization chemistry. The work described here builds on earlier results from the group (ref. 37) consisting in acid-promoted dearomatization of phenylsulfone followed by functionalization of the 6-membered ring with an ester enolate. This manuscript brings this chemistry to the next level by proposing the synthesis of (mainly) fused polycycles which resemble drug molecules. This level of complexity is achieved by the use of reagents tethering two nucleophilic sites, one being an amino group reacting with the ester moiety introduced thanks to the approach developed in ref. 37. This allows the building of a lactam, the second tethered nucleophilic forms another ring after sulfone elimination.

The manuscript reports an important number of new compounds, showing the wide applicability of the described method. The supporting info compiles ^1H and ^{13}C NMR spectra for all compounds, the majority showing no or low levels of impurities. I regret that vibrational spectroscopy was not reported. Many compounds were crystallographically characterized, leaving no doubt about the formulæ thereof. For organic ones, HRMS are not systematically reported, and when it is the case, the deviation from theory is not given. I advise the authors to carefully proof-read the supporting info to correct formatting mistakes (bold/italic, ml, spaces missing, etc).

I think the work reported here is a remarkable piece of organometallic chemistry with application to the synthesis of complex, drug-like molecules, including in an enantioselective manner. It is sufficiently important to attract the attention of the broad readership of Nat. Commun. However, the authors should address the following points and revise their manuscript accordingly before having this material published:

1. In their introduction, the authors emphasize on the power of their approach to achieve the preparation of complex drug-like molecules. The depiction of specific targets (drugs, API or natural product) would be welcome for the reader to gauge the importance or be inspired by the described method.
2. In fig. 1, the abbreviation Tp appears, although it does not seem to be define (by name and/or structure) anywhere
3. Line 135, "the stereoisomer rel-(S,R,S,R,R)-15 could be separated from the others in pure form via recrystallization". Does the yield of 42% in fig. 2 relates to the isolation of this specific stereomer? In the SI, compound 15 seems to be described as a single stereoisomer as well. The authors should be more precise regarding what they report here.
4. I was a bit confused by Fig. 2A. At first, I could not make any difference between 10 and 11, in particular because the adjacent text does not mention the use of enantiopure complexes, and neither does the supporting information (starting material is 3 and not (R)-3). The same goes for 12. Either the authors decide to move the examples with enantiopure complexes to a further section, or explain at this point that they also carried out the syntheses of the 1,2-propanediamine derivatives from (R)- and (S)-3.
5. In fig. 2C, I would not call 5 an impurity in the preparation of 24, but a side-product instead.
6. Line 170, one can read "Calculations support the observation that the bridged tricyclics may be kinetically competitive" but the calculations comparing the kinetic barriers of the bridged vs fused ring formation have not been included. If thermochemical calculations are indeed pertinent to judge the relative stability of isomers I and III, the theoretical confirmation of a favorable pathway can only come by comparing kinetic profiles. From this perspective, I find the mechanistic discussion too light. The same comment applies to the subsequent statements "There is no experimental indication of a purported dienamide [...] than the diene IX" and "Apparently, the kinetic barrier required for the allyl species IIID to bring the tethered amino group to C3, syn to the metal (VI) is too high to be competitive with the addition anti to the metal"
7. The section starting line 174 and finishing line 181 "Interestingly [...] experimentally" is very confusing. First, diene 28 may resemble structure IX but these are not the same compound (Nu3 = CH₂CH₂NH₂ or NH₂, respectively) so the comparison of relative energies with 5 and 24 does not stand. The trans-fused isomer of 5 evoked in line 179-180 should correspond to structure VI in fig. 3 but connection is not made in the text at this point (but later, line 182).
8. Fig. 3 would benefit from inclusion of the calculated relative energies of the isomers VI, VII, VIII and IX (and, as suggested above, kinetic barriers). In addition, to better visualize unbalanced equilibria between IP and ID and IIIP and IIID, equilibrium half-arrows should be of different lengths.
9. Line 191 "Initially, the results were puzzling". Chemistry without puzzling results would not be fun... The authors should stick to the facts and keep their feelings for themselves.
10. Fig. 4A, "or" should be replaced by "and/or". Can approximated yields be given for 29 and 30, or at least ratios vs 31/32?
11. Line 200, "Compound" should not be capitalized
12. Line 201, I think "methine" is incorrect and should be replaced with "methylene"
13. Line 213-214, "followed by the addition of a second arylamine (XII, Fig. 4B). Loss of the labile distal arylamine ligand in XII generates allyl XIII-D" I would like the authors bring more convincing arguments regarding the formation of XII as these types of structures are supposed to be stable product if one refers to their JACS 2022, 144, 9489 paper (ref. 37). Calculations or kinetic data to probe the order in nucleophile would perhaps help?
14. Line 216, followed sounds more natural to me than "proceeded"
15. Line 221-222, "Where Nu2 is S, the reaction proceeds to bicyclic compounds of type XIV"
16. Fig. 4, does the apparently low yield of 34 is the result of the presence of a compound of the type 29? Because of the resemblance of 34 with 31, the fact that the reaction is selective deserves a comment. Moreover, the trend highlighted in the box of fig. 4B where it is stated that structures of the type XIV are only observed when Nu2 is S is wrong, Fig. 4A is full of counter examples. Please clarify.
17. line 223, cysteamine should be connected to compound 36 in the text to improve clarity
18. line 228, description of the exptl conditions to attempt the synthesis of 82 are missing in the SI
19. In the section "Selective formation of complex diastereomers", finally the reader gets answers to his interrogations regarding compounds 11 and 12 in fig. 2A. This should be moved either above, or the enantioenriched examples detailed at this stage. Otherwise, this demonstrates the value of the method by controlling 3 contiguous stereocenters, which is one of the strong points of this work.
20. Line 282, I found the name mobulamide funny and well found. On first reading, I thought it was a true natural product that often get trivial names (in general in link with the organisms they were extracted from). To lift any confusion, I advise the authors to rephrase this sentence and to remove the name from fig. 6B.
21. Similar to my comment about the introduction, in Fig. 6 as well as in the last paragraph, the authors discuss the potential of their method for the synthesis of drugs, and name a number of pharmaceuticals. I believe the impact of the method would be better illustrated by a couple (or more) examples of specific drug/natural product molecules that show cyclic cores matching those reachable by the author's method.

Reviewer #3

(Remarks to the Author)

The manuscript presents a generalized and modular strategy for synthesizing new classes of saturated polycyclic compounds through the use of phenylsulfone tungsten complexes in reactions with various nucleophiles. These structurally diverse polycyclic frameworks hold potential in drug discovery. The comprehensive and compelling findings significantly contribute to the development of polycyclic architectures via benzene dearomatization. I recommend that this promising

study be considered for publication in Nature Communications, contingent upon addressing the following revisions:

1. The chemical identity and structure of the Tp ligand in $Wp(NO)(PMe_3)$ should be clearly defined and described.
2. Two-dimensional NMR spectra should be included in the Supporting Information to fully support the structural assignments.
3. While tungsten complexes are used as key precursors, other transition metal complexes (e.g., Os, Re, Mo) have also been employed in benzene dearomatization. The authors should highlight the unique advantages or reactivity features of tungsten in this context.
4. The selective formation of a single stereoisomer is critical in drug discovery. The manuscript should elaborate on how the metal center influences stereocontrol. A detailed computational study may help elucidate this aspect.
5. The reported transformations are impressive in terms of yield and scope. However, the authors should discuss whether these reactions could be developed into catalytic processes, possibly incorporating chiral ligands to induce enantioselectivity.
6. Since the authors propose this methodology as a potential tool for drug discovery, it would strengthen the manuscript to demonstrate its applicability by synthesizing established pharmaceutical agents using this approach.
7. For geometry optimizations, the use of the 6-31G** basis set (with LANL2DZ for tungsten) is acceptable. However, for more accurate energy evaluations, a higher-level basis set such as def2-TZVP is recommended. Additionally, Grimme's D3(BJ) dispersion correction should be included to account for non-covalent interactions.

Reviewer #4

(Remarks to the Author)

This manuscript reports the synthesis of structurally complex, polycyclic compounds using a tungsten-mediated dearomatization strategy. The work addresses a challenging synthetic target and successfully delivers unique molecular scaffolds with potential relevance to drug discovery. I believe the manuscript is suitable for publication after minor revision. Below are some specific suggestions regarding the computational (DFT) aspects:

1. Currently, only two computational data points are presented in Figure 3. Including the calculated energies for additional structures would make the figure more informative and enhance the reader's understanding of the reaction pathway and stability trends.
2. There is some inconsistency in the nomenclature used. For example, structure VIII appears as a general formula with "Nu3" in the ChemDraw image, but in the Supporting Information, VIII corresponds to a specific structure where Nu3 is NH. Clarifying this point and maintaining consistent labeling throughout the manuscript and SI would help avoid confusion.
3. Please ensure the SI clearly states whether all optimized structures correspond to minima (i.e., no imaginary frequencies).
4. Appropriate references for the computational methods should be cited.
5. In the Supporting Information, the number of decimal places used for energy values is unnecessarily high. For consistency with the main text, please limit energy values to one decimal place.
6. Could the authors comment on why the dienamide isomer of type V is not observed experimentally, while the diene IX, which is calculated to be 3 kcal/mol higher in energy, is formed? This seems counterintuitive and would benefit from further explanation—either thermodynamic or kinetic—to help the reader better understand the observed selectivity.

Version 1:

Reviewer comments:

Reviewer #1

(Remarks to the Author)

The authors addressed my comments in a satisfactory fashion. The manuscript is ready to go.

Reviewer #2

(Remarks to the Author)

The authors have revised their manuscript according to the comments of all the reviewers. I am now satisfied with the current version.

I have picked some typos along my reading of the revised version:

- p2, introduction, 4th paragraph, I think "electron density flows from the metal into the arene π system" is not exact, It should be π^*
- p4, 1st paragraph, "Specifically, Compound 10 was synthesized" compound should not be capitalized
- p4, 2nd paragraph, NOSEY should be replaced by NOESY

-in figure 2, the authors should consider depicting the molecular structure in a nicer rendering than capped sticks. Ellipsoids are probably better suited (with indication of probability level)

-p9, 2nd paragraph "Parenthetically, the lack of an o-phenylenediamine lactam" "o" should be italicized

Reviewer #3

(Remarks to the Author)

The authors have provided detailed and satisfactory responses to the questions I raised. After the revisions, the quality of the manuscript has improved significantly, and it is now suitable for publication in Nature Communications.

Reviewer #4

(Remarks to the Author)

I have reviewed the revised version of the manuscript and the authors' responses to my previous comments. I am pleased to note that all the questions and concerns I raised have been adequately addressed with thoughtful revisions and clear explanations. The modifications made have significantly improved the quality and clarity of the manuscript.

Given the satisfactory responses and the enhanced quality of the paper, I recommend that this manuscript be accepted for publication in Nature Communications.

Responses to reviewer comments: NCOMMS-25-27685-T

Reviewer #1 (Remarks to the Author):

The manuscript by Harman and coworkers reports a new strategy for the synthesis of heteropolycyclic molecules from phenylsulfone which is coordinated to a tungsten complex. This dearomatization allows for a stepwise functionalization. Finally the tungsten fragment is oxidatively removed. This is a refreshing and elegant way to derive at complicated polycyclic structures which are natural-product-like and therefore attractive for drug design and discovery. I like this approach and think that it is suitable for publication in Nature Communications.

A few points should be addressed:

-Abstract and Summary should be improved: It needs to include some more information about the new method.

The abstract and the summary of the original manuscript have been combined into an expanded abstract in the revised manuscript, conforming to the guidelines of Nature Communications. Additional information about the study is also included.

-Introduction or first part of the results section: A comment on the reason why tungsten is especially suitable for this approach would be useful.

A detailed description of the advantages of tungsten has been added and referenced. (lines 63-67)

-Most reader will not be familiar with the employed tungsten-eta²-benzene complexes: Can these complexes be chromatographed like standard organic compounds? A remark would be useful.

A remark has been added confirming that these compounds can be chromatographed and further, that they tolerate exposure to water and air (lines 85-6)

-Figure 2: Reaction conditions for the ester enolate addition should be included.

These conditions have been added to Fig 2.

-Aspects of regioselectivity and diastereoselectivity are not discussed sufficiently (except for mentioning diastereoselectivity for cpds 48 and 49). This is an important point that should be addressed.

We have added several comments concerning regio- and diastereoselectivity at various points in the paper and these have been highlighted.

-Compounds 48 and 49: The authors state that proton NMR data confirm the presence of just one diastereomer. But what is their enantiomeric excess?

According to the supplier the (2R)-Propane-1,2-diamine and (2S)-Propane-1,2-diamine are >99% ee. As such, the ee of the observed complexes **11** and **12** and their organics **48** and **49**, respectively, should also be >99% ee. This is now indicated in the captions to Fig 2 and Fig 5.

-Mechanistic Figure 3: The authors write that the allyl complexes IP and ID differ in their conformations. That appears to be wrong. IP and ID are not conformers but instead may be called regioisomers. By the way: Is it possible that IP and ID are not in equilibrium but rather resonance structures and exist as an eta³-allyl complex? Same for IIID and IIIP.

IP and ID are indeed conformers. Rotational conformers are the most common type, but conformers can involve inversion of a bond angle (akamptisomers), trigonal pyramidal inversion, or other rearrangements. Two references (40, 41) has been added to clarify (justify) the usage of the term 'conformer'. The key point here is that the IP and ID are two distinct forms of the η^2 allyl complexes, they are separated by a low-energy barrier (a few kcal, similar to rotational conformers) and they structurally differ primarily by their bond and dihedral angles. Note that both P and D conformers have three W-C bonds (hence we do not like the term linkage isomer or constitutional isomer in this application), they just differ by length (as explained in Reference 40).

Reviewer #2 (Remarks to the Author):

This article from Siano et al. reports on the derivatization of phenylsulfone coordinated in an η^2 fashion to a Tp-supported nitrosyl tungsten complex. Binding to tungsten ensure sufficient electron-richness to the arene so that it can be easily protonated, the entry point for dearomatization and functionalization chemistry. The work described here builds on earlier results from the group (ref. 37) consisting in acid-promoted dearomatization of phenylsulfone followed by functionalization of the 6-membered ring with an ester enolate. This manuscript brings this chemistry to the next level by proposing the synthesis of (mainly) fused polycycles which resemble drug molecules. This level of complexity is achieved by the use of reagents tethering two nucleophilic sites, one being an amino group reacting with the ester moiety introduced thanks to the approach developed in ref. 37. This allows the building of a lactam, the second tethered nucleophilic forms another ring after sulfone elimination.

The manuscript reports an important number of new compounds, showing the wide applicability of the described method. The supporting info compiles ^1H and ^{13}C NMR spectra for all compounds, the majority showing no or low levels of impurities. I regret that vibrational spectroscopy was not reported. Many compounds were crystallographically characterized, leaving no doubt about the formulæ thereof.

We usually take IR spectra in cases where the spectrum can help uniquely identify the compound. Usually this means a functional group (e.g., amide, nitrile, nitro) that we have not determined conclusively through X-ray or CNMR. That said, in the spirit of the reviewer's request, for compounds what we did not have to remake, we have recorded IR spectra and include their key features in the SI. Compounds that already have listed IR data: 5, 16, 23, 29, 31. IR data added for compounds: 7, 70, 75, 79-81, and 83.

For organic ones, HRMS are not systematically reported, and when it is the case, the deviation from theory is not given. I advise the authors to carefully proof-read the supporting info to correct formatting mistakes (bold/italic, ml, spaces missing, etc).

We have provided HRMS for all compounds in which X-ray data was unavailable. According to Nature Comm. Guidelines, the deviation is not to be reported. We have modified the formatting of HRMS reports to better match Nature guidelines and to be consistent throughout.

I think the work reported here is a remarkable piece of organometallic chemistry with application to the synthesis of complex, drug-like molecules, including in an enantioselective manner. It is sufficiently important to attract the attention of the broad readership of Nat. Commun. However, the authors should address the following points and revise their manuscript accordingly before having this material published:

1. In their introduction, the authors emphasize on the power of their approach to achieve the preparation of complex drug-like molecules. The depiction of specific targets (drugs, API or natural product) would be welcome for the reader to gauge the importance or be inspired by the described method.

The message that we are trying to convey is access to NEW chemical space. In this sense, there are no targets. The goal is a broad modular approach to multicyclic cores that have not previously been

accessible, but that have properties that make them interesting candidates for small molecule drugs. We have tried to better communicate this in the rewritten introduction.

2. In fig. 1, the abbreviation Tp appears, although it does not seem to be define (by name and/or structure) anywhere

Tp has been defined (line 66 and Fig 1 caption) and the structure included in Fig. 2.

3. Line 135, “the stereoisomer rel-(S,R,S,R,R)-15 could be separated from the others in pure form via recrystallization”. Does the yield of 42% in fig. 2 relates to the isolation of this specific stereomer? In the SI, compound 15 seems to be described as a single stereoisomer as well. The authors should be more precise regarding what they report here.

Thank you for pointing out this error. The main text (lines 138-147) and Fig 2 , as well as the SI has been modified to better describe the situation: “In the case of $\text{HNu}_2 - \text{Nu}_3\text{H} = (S,S)\text{-trans-1,2-}$ diaminocyclohexane, $(S_w,3R,4S,5R,8S,9S)\text{-15}$ was isolated via chromatography in 42% yield, this compound appearing in the methanol fraction. Since **15** was obtained using a racemic form of phenylsulfone complex **1**, only 50% of **1** was available to form this stereoisomer. Of note, the other possible stereoisomer of **15**, $(R_w,3R,4S,5R,8S,9S)\text{-15}$, was never identified. Instead, the other hand of the metal was isolated as $(R_w,3S,4R,5S,8S,9S)\text{-41}$ (vide infra, Fig. 4), which crystallized (35% yield out of a theoretical yield of 50%) from the low-polarity fractions of the chromatography procedure for **15**. The ee for compounds **15** and **41** is determined by the optical purity of the diamine, and the Flack parameter for the corresponding crystal structures confirms the absolute stereochemistry shown in Fig 2. and Fig 4.” In addition to this, Fig 2 was modified to include the molecular structure of **15** (XC-SRD).

4. I was a bit confused by Fig. 2A. At first, I could not make any difference between 10 and 11, in particular because the adjacent text does not mention the use of enantiopure complexes, and neither does the supporting information (starting material is 3 and not (R)-3). The same goes for 12. Either the authors decide to move the examples with enantiopure complexes to a further section, or explain at this point that they also carried out the syntheses of the 1,2-propanediamine derivatives from (R)- and (S)-3.

This ambiguity has been addressed in the main text (lines 120-126): “When the incorporated diamine itself contained additional stereocenters, diastereomers resulted. Specifically, Compound **10** was synthesized as a mixture of diastereomers starting from racemic **3** and (R)-propane-1,2-diamine. In contrast, compound **11** and **12** were prepared using enantiopure (R)-**3**, in combination with (R)-propane-1,2-diamine or (S)-propane-1,2-diamine, respectively (vide infra).”

5. In fig. 2C, I would not call 5 an impurity in the preparation of 24, but a side-product instead.

This has been corrected in Fig 2.

6. Line 170, one can read “Calculations support the observation that the bridged tricyclics may be kinetically competitive” but the calculations comparing the kinetic barriers of the bridged vs fused ring formation have not been included. If thermochemical calculations are indeed pertinent to judge the relative stability of isomers I and III, the theoretical confirmation of a favorable pathway can only come by comparing kinetic profiles. From this perspective, I find the mechanistic discussion too light. The same comment applies to the subsequent statements “There is no experimental indication of a purported dienamide [...] than the diene IX” and “Apparently, the kinetic barrier required for the allyl species IIID to bring the tethered amino group to C3, syn to the metal (VI) is too high to be competitive with the addition anti to the metal”

As the reviewer indicates, we did not carry out transition state calculations, partly because the mechanisms involve proton transfers that are not well defined, and we don't want to unnecessarily speculate.

We have reworded this section, including:

While the bridged tricyclics may be kinetically competitive, calculations indicate that they are strongly disfavored thermodynamically compared to the fused tricyclics (lines 173-6).

But we stand by the original wording of:

“There is no experimental indication of a purported dienamide isomer of type **V**, even though DFT calculations place such a species about 3 kcal/mol lower in energy than the diene **IX** (189-197)” and: “Apparently, the kinetic barrier required for the allyl species **IIID** to bring the tethered amino group to C3, syn to the metal (**VI**) is too high to be competitive with the addition anti to the metal (199-201).” Even without transition state calculations we can make these observations.

7. The section starting line 174 and finishing line 181 “Interestingly [...] experimentally” is very confusing. First, diene 28 may resemble structure IX but these are not the same compound (Nu3 = CH₂CH₂NH₂ or NH₂, respectively) so the comparison of relative energies with 5 and 24 does not stand. The trans-fused isomer of 5 evoked in line 179-180 should correspond to structure VI in fig. 3 but connection is not made in the text at this point (but later, line 182).

Excellent point made by this reviewer. We have reworded this section (lines 189-197) and have added energies for the case where HNu₃ = NH₂ to Fig. 3.

8. Fig. 3 would benefit from inclusion of the calculated relative energies of the isomers VI, VII, VIII and IX (and, as suggested above, kinetic barriers). In addition, to better visualize unbalanced equilibria between IP and ID and IIP and IIID, equilibrium half-arrows should be of different lengths.

This has been done in Fig. 3.

9. Line 191 “Initially, the results were puzzling”. Chemistry without puzzling results would not be fun... The authors should stick to the facts and keep their feelings for themselves.

Removed, but only under protest 😊

10. Fig. 4A, “or” should be replaced by “and/or”. Can approximated yields be given for 29 and 30, or at least ratios vs 31/32?

The figure has been modified. We do not have the data available that would allow us to report a ratio (from the reaction mixture). We are reporting isolated yields.

11. Line 200, “Compound” should not be capitalized.

corrected

12. Line 201, I think “methine” is incorrect and should be replaced with “methylene”

corrected (depending on compound it is either methine or methylene)

13. Line 213-214, “followed by the addition of a second arylamine (XII, Fig. 4B). Loss of the labile distal arylamine ligand in XII generates allyl XIII-D” I would like the authors bring more convincing arguments regarding the formation of XII as these types of structures are supposed to be stable product if one refers to their JACS 2022, 144, 9489 paper (ref. 37). Calculations or kinetic data to probe the order in nucleophile would perhaps help?

“While allylic amine complexes are stable under basic conditions, in the presence of weak acids (i.e. ammonium salts in the reaction mixture) they are labile.” We added this statement and provided a reference supporting the claim (ref 42; Wilson et al 2021, Helvetica Chimica Acta). At this point, we do not have the resources/manpower to carry out the suggested kinetics experiments, but the mechanism we propose is logical and is supported by the data we have collected.

Line 216, followed sounds more natural to me than “proceeded”

Corrected.

Line 221-222, “Where Nu₂ is S, the reaction proceeds to bicyclic compounds of type XIV”

added “the” before “type”.

16. Fig. 4, does the apparently low yield of **34** is the result of the presence of a compound of the type **29**? Because of the resemblance of **34** with **31**, the fact that the reaction is selective deserves a comment. Moreover, the trend highlighted in the box of fig. 4B where it is stated that structures of the type XIV are only observed when Nu₂ is S is wrong, Fig. 4A is full of counter examples. Please clarify.

The low yield of **34** is due to the lower reactivity of diaminonaphthalene. This leads to partial decomposition. The other isomer (**29**) is not observed in this reaction. We added a sentence to help clarify this situation. Fig 4B has been corrected. Thank you for catching this error.

17. line 223, cysteamine should be connected to compound **36** in the text to improve clarity

Addressed in line 247.

18. line 228, description of the exptl conditions to attempt the synthesis of **82** are missing in the SI.

Experimental conditions for **82** were added to the SI as follows:

Compound **2** (100 mg, 0.136 mmol) was placed in a test tube with ACN (2 mL), and chilled to -30 °C. After 10 min, a 1 M HOTf/ACN (0.273 mL, 0.273 mmol) solution was added to the test tube and the solution was allowed to stir at -30 °C for 30 min. In a separate test tube, aniline (0.123 mL, 1.360 mmol) with ACN (2 mL) was cooled at -30 °C for 20 min. After the time elapsed, the former solution was added to the latter, dropwise. The reaction stirred at -30 °C for 48 h and room temperature for 5 h. The organic layer was evaporated in vacuo and concentrated to a thin yellow film. The resulting film was dissolved in minimal DCM and pipetted into 25 mL of stirring hexane. An off-white solid precipitated out and was collected on a 15 mL fine porosity fitted disk, washed with hexane (2 × 10 mL) and desiccated overnight to yield compound **82** (97 mg, 0.116 mmol, 85%).

19. In the section “Selective formation of complex diastereomers”, finally the reader gets answers to his interrogations regarding compounds **11** and **12** in fig. 2A. This should be moved either above, or the enantioenriched examples detailed at this stage. Otherwise, this demonstrates the value of the method by controlling 3 contiguous stereocenters, which is one of the strong points of this work.

We hope the rewrite of lines 120-126 (see above) solves this problem, but just in case we added a *vide infra* on line 126, indicating to the reader that we will return the subject of enantioenriched compounds.

20. Line 282, I found the name mobulamide funny and well found. On first reading, I thought it was a true natural product that often get trivial names (in general in link with the organisms they were extracted from). To lift any confusion, I advise the authors to rephrase this sentence and to remove the name from fig. 6B.

We have rephrased the description of the naming of this compound (lines 309-310) to make clear that it is not a natural product and we have put quotes around the name in Fig 6.

21. Similar to my comment about the introduction, in Fig. 6 as well as in the last paragraph, the authors discuss the potential of their method for the synthesis of drugs, and name a number of pharmaceuticals. I believe the impact of the method would be better illustrated by a couple (or more) examples of specific drug/natural product molecules that show cyclic cores matching those reachable by the author's method.

We have added the following sentences (lines 348-350): Specifically, we recently synthesized γ -lycorane from the tungsten benzene complex in 6 steps (25% overall yield).⁴² However, the polycyclic cores that have not yet been prepared in a lab or found in nature are the most exciting potential targets for this methodology.

Reviewer #3 (Remarks to the Author):

The manuscript presents a generalized and modular strategy for synthesizing new classes of saturated polycyclic compounds through the use of phenylsulfone tungsten complexes in reactions with various nucleophiles. These structurally diverse polycyclic frameworks hold potential in drug discovery. The comprehensive and compelling findings significantly contribute to the development of polycyclic architectures via benzene dearomatization. I recommend that this promising study be considered for publication in Nature Communications, contingent upon addressing the following revisions:

The chemical identity and structure of the Tp ligand in $W Tp(NO)(PMe_3)$ should be clearly defined and described.

Now defined on line 66

2. Two-dimensional NMR spectra should be included in the Supporting Information to fully support the structural assignments.

COSY, NOSEY, HSQC, and HMBC spectra for select representative compounds (**11**, **12**, **23**, **29/31**, **50**, **65**, **81**, and **85**) have been added to the SI. These compounds were chosen to have at least one example of each compound type from the various sections of the maintext. Key NOE interactions were also highlighted for **11**, **12**, and **81**, to help emphasize especially interesting structural features. Please keep in mind we have 35 crystal structures supporting our claims as well.

3. While tungsten complexes are used as key precursors, other transition metal complexes (e.g., Os, Re, Mo) have also been employed in benzene dearomatization. The authors should highlight the unique advantages or reactivity features of tungsten in this context.

Now addressed on lines 64-67

4. The selective formation of a single stereoisomer is critical in drug discovery. The manuscript should elaborate on how the metal center influences stereocontrol. A detailed computational study may help elucidate this aspect.

Explanation added in lines 120-122 and again in 170-171.

5. The reported transformations are impressive in terms of yield and scope. However, the authors should discuss whether these reactions could be developed into catalytic processes, possibly incorporating chiral ligands to induce enantioselectivity.

The challenge of running these reactions catalytically is discussed in detail in our review (reference 16).

6. Since the authors propose this methodology as a potential tool for drug discovery, it would strengthen the manuscript to demonstrate its applicability by synthesizing established pharmaceutical agents using this approach.

We have added the following sentences (lines 348-350): Specifically, we recently synthesized γ -lycorane from the tungsten benzene complex in 6 steps (25% overall yield).⁴² However, the polycyclic cores that have not yet been prepared in a lab or found in nature are the most exciting potential targets for this methodology.

7. For geometry optimizations, the use of the 6-31G** basis set (with LANL2DZ for tungsten) is acceptable. However, for more accurate energy evaluations, a higher-level basis set such as def2-TZVP is recommended. Additionally, Grimme's D3(BJ) dispersion correction should be included to account for non-covalent interactions.

Benchmarking studies for similar tungsten complexes were demonstrated by Harman and coworkers. The M06/6-31G** (with LANL2DZ for tungsten) methodology was emphasized for its excellent agreement to SC-XRD structures and accurate relative energies in comparison to experimental values. Comparisons between M06/Def2-TZVPD and M06/6-31G** (with LANL2DZ for tungsten) showed a higher-level basis set did not significantly improve energy evaluations.

Additionally, the inclusion of Grimme's dispersion (D3) is generally recommended, even for Minnesota functionals, but a study by Grimme and coworkers indicates that "M06, MN15L, and MN15 are the only Minnesota DFAs that do not seem to benefit from the DFT-D3 correction."

Smith, J.A., Schouten, A., Wilde, J.H., Westendorff, K.S., Dickie, D.A., Ess, D.H., Harman, W.D. Experiments and Direct Dynamics Simulations That Probe η^2 -Arene/Aryl Hydride Equilibria of Tungsten Benzene Complexes. *J. Am. Chem. Soc.* **2020**, 142, 38, 16437–16454.

Goerigk, L., Hansen, A., Bauer, C., Ehrlich, S., Najibi, A., Grimme, S. A look at the density functional theory zoo with the advanced GMTKN55 database for general main group thermochemistry, kinetics and noncovalent interactions. *Phys. Chem. Chem. Phys.*, **2017**, **19**, 32184-32215.

Reviewer #4 (Remarks to the Author):

This manuscript reports the synthesis of structurally complex, polycyclic compounds using a tungsten-mediated dearomatization strategy. The work addresses a challenging synthetic target and successfully delivers unique molecular scaffolds with potential relevance to drug discovery. I believe the manuscript is suitable for publication after minor revision. Below are some specific suggestions regarding the computational (DFT) aspects:

1. Currently, only two computational data points are presented in Figure 3. Including the calculated energies for additional structures would make the figure more informative and enhance the reader's understanding of the reaction pathway and stability trends.

Figure 3 has been modified to include energies of four different isomers. We did not carry out transition state calculations, partly because the mechanisms involve proton transfers that are not well defined, and we don't want to unnecessarily speculate.

2. There is some inconsistency in the nomenclature used. For example, structure VIII appears as a general formula with "Nu3" in the ChemDraw image, but in the Supporting Information, VIII corresponds to a specific structure where Nu3 is NH. Clarifying this point and maintaining consistent labeling throughout the manuscript and SI would help avoid confusion.

The figure in the SI was updated to be more consistent with the main text and Fig. 3. Nu² and Nu³ are now defined as anions, where HNu—NuH would represent a neutral diamine.

3. Please ensure the SI clearly states whether all optimized structures correspond to minima (i.e., no imaginary frequencies).

Text at the beginning of the DFT section in the SI was modified from:

“Ground-state structures were optimized at the M06 level of theory using the 6-31G** [LANL2DZ for W] basis set in Gaussian 16. Previous literature demonstrates that this functional and basis set choice accurately corroborates experimental results. Solvent effects of acetonitrile were modeled using SMD. Gaussian’s default criteria were used for optimization, **vibrational frequency analysis verified that structures were minima** and thermal free energy corrections were applied.

To:

“Ground-state structures were optimized at the M06 level of theory using the 6-31G** [LANL2DZ for W] basis set in Gaussian 16. Previous literature demonstrates that this functional and basis set choice accurately corroborates experimental results. Solvent effects of acetonitrile were modeled using SMD. Gaussian’s default criteria were used for optimization, **vibrational frequency analysis verified that all ground state structures contain no imaginary frequencies**, and thermal free energy corrections were applied.”

4. Appropriate references for the computational methods should be cited.
References were added to the SI (2 and 3) for the computational methods.

5. In the Supporting Information, the number of decimal places used for energy values is unnecessarily high. For consistency with the main text, please limit energy values to one decimal place.

All energy values, in both the table and figures, were rounded to one decimal place.

6. Could the authors comment on why the dienamide isomer of type V is not observed experimentally, while the diene IX, which is calculated to be 3 kcal/mol higher in energy, is formed? This seems counterintuitive and would benefit from further explanation—either thermodynamic or kinetic—to help the reader better understand the observed selectivity.

Added to text (lines 194-197): We speculate that this is a kinetic phenomenon in which deprotonation at the bridgehead carbon (C4) is challenging, in part because such a reaction would require a base to approach from the same side as the sterically demanding metal complex.